# Antibiotic Prescribing Quality in Out-of-Hours Primary Care and Critical Appraisal of Disease-Specific Quality Indicators

**DOI:** 10.3390/antibiotics8020079

**Published:** 2019-06-12

**Authors:** Annelies Colliers, Niels Adriaenssens, Sibyl Anthierens, Stephaan Bartholomeeusen, Hilde Philips, Roy Remmen, Samuel Coenen

**Affiliations:** 1Department of Primary and Interdisciplinary Care (ELIZA)—Centre for General Practice, Faculty of Medicine and Health Sciences, University of Antwerp, Doornstraat 331, B-2610 Antwerp, Belgium; niels.adriaenssens@uantwerpen.be (N.A.); sibyl.anthierens@uantwerpen.be (S.A.); stephaan.bartholomeeusen@uantwerpen.be (S.B.); hilde.philips@uantwerpen.be (H.P.); roy.remmen@uantwerpen.be (R.R.); samuel.coenen@uantwerpen.be (S.C.); 2Department of Epidemiology and Social Medicine (ESOC), Faculty of Medicine and Health Sciences, University of Antwerp, Universiteitsplein 1, B-2610 Antwerp, Belgium; 3Vaccine & Infectious Disease Institute (VAXINFECTIO), Faculty of Medicine and Health Sciences, University of Antwerp, Universiteitsplein 1, B-2610 Antwerp, Belgium

**Keywords:** antibiotics, out-of-hours care, primary care, quality of care, quality indicators, practitioners cooperative

## Abstract

Outpatient antibiotic use in Belgium is among the highest in Europe. The most common reason for an encounter in out-of-hours (OOH) primary care is an infection. In this study, we assessed all consultations from July 2016 to June 2018 at five OOH services. We described antibiotic prescribing by diagnosis, calculated disease-specific antibiotic prescribing quality indicators’ (APQI) values and critically appraised these APQI. We determined that 111,600 encounters resulted in 26,436 (23.7%) antibiotic prescriptions. The APQI diagnoses (i.e., bronchitis, upper respiratory infection, cystitis, tonsillitis, sinusitis, otitis media, and pneumonia) covered 14,927 (56.7%) antibiotic prescriptions. Erysipelas (1344 (5.1%)) and teeth/gum disease (982 (3.7%)) covered more prescriptions than sinusitis or pneumonia. Over 75% of patients with tonsillitis and over 50% with bronchitis, sinusitis, and otitis media were prescribed an antibiotic. Only for otitis media the choice of antibiotic was near the acceptable range. Over 10% of patients with bronchitis or pneumonia and over 25% of female patients with an acute cystitis received quinolones. The APQI cover the diagnoses for only 57% of all antibiotic prescriptions. As 5.1% and 3.7% of antibiotic prescriptions are made for erysipelas and teeth/gum disease, respectively, we propose to add these indications when assessing antibiotic prescribing quality in OOH primary care.

## 1. Introduction

Antibiotic use in Belgium is among the highest in Europe, especially in ambulatory care, and thus the risk for antimicrobial resistance is high [1]. The European Surveillance of Antimicrobial Consumption (ESAC) [2] project has developed a set of 21 disease-specific antibiotic prescribing quality indicators (APQI) to assess the quality of antibiotic prescribing in primary care [3]. These APQI have been adopted in several evaluations of antibiotic prescribing in Europe [4,5,6]. 

Out-of-hours (OOH) care covers a large part of hours during a week. In Belgium, but also in the rest of Europe, the on-going establishment of large-scale general practitioner cooperatives (GPCs) represents one of the most important developments for primary OOH health care. The most common reason for an encounter in Flemish (Northern part of Belgium) OOH primary care is an infection [7]. Many infections are self-limiting and no antimicrobial treatment is necessary. However, in our qualitative work general practitioners (GPs) describe the difficulties they encounter in OOH care not to prescribe antibiotics relating to contextual factors such as a larger uncertainty due to an unknown patient, type of patients (e.g., children, elderly, Non-Native Speakers, …), workload, and the lack of diagnostic tools or follow-up. They have a feeling that their professional identity is different compared to in-hours care [8,9]. However, previous research in Belgium showed similar APQI values for in- and out-of-hours care, using the data of a singular general practitioner cooperative (GPC) from 2004–2009 [4], which is similar to findings in Norway and Sweden [10,11]. In the Netherlands, research showed higher prescribing rates in OOH care but with equal or even better quality compared to in-hours care [12]. In the United Kingdom, researchers found an increase in antibiotic prescribing in OOH primary care in contrast to in-hours care, however, this study did not explore the quality of the prescriptions [13,14]. Moreover, in Danish OOH care, antibiotics are the most prescribed drugs, but there was no evaluation on the quality of prescribing [15].

Therefore, the objective of this study is to assess the quality of antibiotic prescribing in OOH primary care to inform the implementation of future interventions to improve the antibiotic prescribing quality in this specific setting [16]. More specifically, we aim to describe antibiotic prescribing in Belgian OOH primary care by indication, assess its quality by updating values for ESAC’s disease-specific APQI [3] and critically appraise these APQI [4].

## 2. Methods

### 2.1. Database and GPCs

Data were extracted from the Improving Care and Research Electronic Data Trust Antwerp (iCAREdata) database, a research database of linked data on OOH primary care [17,18]. On a weekly basis, routine clinical data and patient characteristics are automatically collected from GPCs, emergency departments (EDs) and pharmacists’ electronic records completed during OOH-care. For this study we only used data from GPCs.

iCAREdata currently covers nine GPCs, that cover a population of 1,127,153. Only data from five GPCs were withheld for this study, because of the level of completeness of their data and availability at the start of the study. Together, they cover a population of around 811,000 inhabitants (11% of the Flemish population; i.e., the Northern Dutch speaking part of Belgium). The characteristics of these five GPCs are presented in Table 1. Because of a lack of registration of home visits in OOH care, only data from consultations at the GPC itself were used. 

### 2.2. Study Setting

Since the beginning of the year 2000, Belgian OOH care is increasingly being organized in GPCs in regions that typically cover 80–180 GPs. All practicing GPs in such an area are obliged to participate in this service following a rotation-based system of being ‘on call’ during the weekends in their own region and they work in shifts of on average 12 h. GPCs are mostly organized in a fee-for-service system. There are 80–180 GPs per GPC. Belgian GPs do not have a gatekeeper function. There is free access to primary, secondary, and tertiary care. They use an electronic medical health record to register every patient contact. The software can differ depending on the GPC, but diagnoses are always selected using Thesaurus terms in the Belgian Bilingual Biclassified Thesaurus (3BT). By selecting a diagnosis in 3BT, an International Classification of Primary Care (ICPC-2-R) code is automatically linked [19]. Prescriptions are registered using the database of the Belgian Centre for Pharmacotherapeutic Information [20], which is linked to the Anatomical Therapeutic Chemical (ATC) classification system [21,22].

### 2.3. Data Analysis

Data extraction was done using Microsoft SQL server 2012. We performed descriptive analyses using Microsoft Excel 2016. Data from 1 July 2016 until 30 June 2018 were included. 

#### 2.3.1. Antibiotic Prescribing by Indication

To describe antibiotic prescribing by indication, first we only analyzed all antibiotic prescriptions (ATC code J01: antibacterials for systemic use) delivered during the consultations in five GPCs during a two-year timeframe and linked them with the diagnosis that was registered in the electronic medical health record. 

#### 2.3.2. Quality of Antibiotic Prescribing: Disease-Specific APQI

Second, we used information on all encounters to assess the quality of antibiotic prescribing using the disease-specific APQI introduced by ESAC (European Surveillance of Antimicrobial Consumption project) in 2007 [3]. The seven most common indications for antibiotic prescribing are in descending order: acute bronchitis (ICPC code R78), acute upper respiratory tract infection (RTI) (R74), cystitis/other urinary infection (UTI; U71), acute tonsillitis (R76), acute/chronic sinusitis (R75), and acute otitis media (H71). For pneumonia (R81), values of three valid APQI were calculated: a = the percentage of patients with age and/or gender limitation (see legend Table 3) prescribed an antibiotic;b = a and receiving the guideline recommended antibiotic;c = a and receiving quinolones.

We compared them with the findings of an earlier study with data from 2004–2009 from one GPC [4].

#### 2.3.3. Critical Appraisal

We evaluated if APQI is a sufficient method when describing the quality of antibiotic prescribing in OOH care. We used and adapted the concept of drug utilization 90% (DU90%) [23], which analyzes the number of drugs accounting for 90% of drug use. We used it to see if using APQI-diagnoses covers 90% of all antibiotic prescriptions.

### 2.4. Ethics

The study was approved by the Ethics Committee of the Antwerp University Hospital/University of Antwerp (reference number 17/08/089), and registered at clinicaltrials.gov (NCT03082521). The study was approved by the scientific advisory board of ICAREdata and the different GPCs.

## 3. Results

From 1/7/2016 until 30/6/2018, 26,436 antibiotic prescriptions were registered in 111,600 visits (excluding home visits) covering a population of 811,097 persons. 

For 93.65% of the prescriptions, a diagnosis was registered in the same consultation. A diagnosis was missing for 1612 (6.35%) antibiotic prescriptions.

### 3.1. Antibiotic Prescribing by Indication

Table 2 shows the number of prescriptions linked to the ICPC chapters. Of them, 44% of the prescriptions are linked to diagnoses related to respiratory infections, 13% to urinary tract infections, 12% skin infections, 11% ear infections, and 6% digestive infections. Other diagnoses represent less than 5% of the prescriptions and are grouped in other diagnoses. 

### 3.2. The Quality of Antibiotic Prescribing

Table 3 shows the results of the quality appraisal using APQI. We complemented this table with similar indicators for erysipelas and dental abscess, because they both cover respectively 5% and 4% of antibiotic prescriptions. There are not yet quality indicators for these two indications [24], but we chose to use following age limitations: older than one year for erysipelas and older than 18 year for dental abscess. Overall, for all conditions the percentage of antibiotics prescribed for common conditions exceed the upper limit (i.e., 20% for otitis media, upper RTI, sinusitis and tonsillitis, and 30% for bronchitis). Despite the fact that the highest amount of prescribed antibiotics are linked to upper RTI (Table 3), indicator 2A shows that 30% of patients diagnosed with upper RTI receive a prescription. Of the patients diagnosed with tonsillitis, 77% received an antibiotic prescription. For the latter two conditions, the use of the recommended antibiotics is less than 10% (3% for upper RTI and 6% for tonsillitis). For most other conditions, the use of recommended antibiotics ranges from 40–46% (40% for sinusitis, 42% for bronchitis, 42% for teeth/gum disease, erysipelas, and 46% for pneumonia). For cystitis and otitis media, the use of recommended antibiotics is higher, 69% and 74% respectively. No conditions reached the goal of 80–100%. Comparing these results with the APQI OOH study from 2004–2009, there is an improvement in choosing the recommended antibiotic. (otitis media: 42% → 74%; acute sinusitis: 23% → 40%; cystitis 40% → 69%; bronchitis 34% → 42%). Other values are in the same range as in 2004–2009 (Appendix A).

For four conditions, the 5% upper limit of quinolone use was exceeded, i.e., in ascending order sinusitis (7%), bronchitis (11%), pneumonia (15%), and cystitis (25%). 

GPs can choose to enter a symptom diagnosis. (such as fever, cough, etc.), but only 3.8% of all antibiotic prescriptions are linked to this type of diagnosis (Appendix A).

### 3.3. Critical Appraisal of APQI to Describe Antibiotic Prescribing Quality in OOH Care

In 31,596/111,600 consultations, a diagnosis covered by APQI or erysipelas or dental abscess was made in OOH. Detailed diagnoses linked to antibiotic prescription are shown in Table 4. Most prescriptions are linked to upper respiratory tract infections (13%), followed by prescriptions for acute cystitis (12%), acute tonsillitis (9%) acute bronchitis (8%), and acute otitis media (8%). Most antibiotic prescriptions for skin infections are made for erysipelas (5%). Sinusitis, teeth/gum disease and pneumonia represent 4%, 4%, and 3% of the total of antibiotic prescriptions, respectively. When only using APQI diagnoses, 57% of antibiotic prescriptions are covered. When including erysipelas and teeth/gum disease, these nine diagnoses represent 66% of all antibiotic prescription. DU90% normally describes the number of drugs. When applying this concept on the coverage of indications, we do not reach 90% of all antibiotic prescriptions. The 220 other diagnoses represent in total 28% of the prescriptions but represent less than 3% each.

## 4. Discussion

### 4.1. Main Findings

Although most respiratory tract infections are self-limiting and usually caused by viral pathogens, at least half of the patients diagnosed with acute otitis media, sinusitis, tonsillitis, and bronchitis receive a prescription for an antibiotic at the GPC. Patients diagnosed with acute tonsillitis receive in 77% of the cases an antibiotic. Only for patients diagnosed with acute upper respiratory tract infection is the proportion of antibiotic prescriptions lower, but still higher than acceptable (30% while the acceptable range is between 0–20%). For nearly all indications the recommended type of antibiotics, according to the guidelines, are not prescribed apart from for otitis media, which is almost within the accepted range. Quinolone use is outside the acceptable range for following respiratory infections: sinusitis (7%), bronchitis (11%), and pneumonia (15%), which is in line with previous studies. The highest proportion of quinolones are still found among antibiotics prescribed for female patients with acute cystitis (25%). Looking at the total amount of antibiotic prescribing, we noticed a high number of prescriptions for erysipelas and dental problems, again with a high prescribing rate of non-guideline recommended antibiotics. There is not yet an acceptable range of prescribing determined for these two conditions. We suggest an acceptable range for the recommended antibiotic of more than 80% and respectively >80% and <30% of total antibiotic prescribing for erysipelas and teeth/gum disease.

For acute tonsillitis and acute upper respiratory tract infection, the very low percentage of recommended antibiotics (6% and 3%, respectively, acceptable range 80–100%) can be explained by the frequent unavailability of small spectrum penicillin in the Belgian pharmacies, therefore GPs most often choose to prescribe amoxicillin as an alternative (63% and 66% of all antibiotic prescriptions for upper RTI and tonsillitis respectively are for amoxicillin, see Appendix A) [4]. It is likely that conditions with recommended use around 40–44% (40% for sinusitis, 42% for bronchitis, 43% for teeth/gum disease, and 44% for erysipelas and 46% for pneumonia) are of more interest to policy makers, as for these conditions the GP decides to deviate from the guidelines mostly in favor of a broader spectrum antibiotic e.g., amoxicillin with clavulanic acid instead of amoxicillin alone). More detailed research is needed to identify appropriate use of broader spectrum antibiotic (e.g., treatment failure of initiated small spectrum antibiotic) versus inappropriate use driven by other factors such as lack of knowledge of the guideline, disagreement with the guideline, etc. [8]. These factors have been suggested as drivers for the use of quinolones for respiratory tract infections [25]. 

In Belgium, since 1 May 2018, the conditions for the reimbursement of quinolones and fluoroquinolones have been changed. These antibiotics are no longer reimbursed for the treatment of respiratory tract infections or uncomplicated urinary tract infections. Immediately after 1 May 2018, this measure seemed to have had an effect on quinolone prescribing [26]. However, whether or not this effect is causal and whether this trend will continue will be further monitored and analyzed using iCAREdata [17,18].

We have used the disease-specific APQI to describe the quality of antibiotic prescribing during OOH care. In our study, all seven diagnoses included in these APQI represent a considerable proportion of the total antibiotic use. In addition, we noticed that a substantial number of antibiotic prescriptions was also linked to two other diagnoses, i.e., erysipelas, dental problems (erysipelas (ICPC S76), and teeth/gum disease (ICPC D82), the latter most often representing a dental abscess based on notes review in a sample of the data. Therefore, we propose to add these two diagnoses (ICPC codes) when assessing the quality of antibiotic prescribing in OOH care. Further validation and more studies are necessary to confirm their relevance. However, when using these nine indications, only 66% of antibiotic prescriptions are covered and it could be argued that this is not enough to describe antibiotic quality. In accordance with the DU90% [23] concept, it would be more relevant to use indicators to be able to describe 90% of prescriptions. Dolk et al. found that the majority of antibiotic prescriptions in English primary care were for infections of the respiratory and urinary tracts, followed by skin infections and only a small number for dental problems [27]. 

According to the Belgian antibiotic guidelines, the primary treatment of acute dental problems is dental care by a dentist [28]. However, patients with acute tooth ache often visit the GPC to seek help during weekends and receive antibiotics. This has also been shown in previous studies in the UK [29]. GPs feel that general practice is not the best setting for managing dental problems [30] and one could argue whether or not the care for dental problems falls within the remit of GPs. Access to dental care during the weekends is limited, but patients should be redirected to these services.

APQI’s were calculated for 1 GPC from data from 2004 to 2009. In the current study, with five GPCs (including the one GPC that was used in the 2004–2009 APQI OOH study) we noticed similar results in the percentage of patients who received an antibiotic, but an improvement in choice of antibiotics. 

### 4.2. Strengths and Limitations

We have used electronic data that were routinely registered by GPs in their electronic health record for each patient contact. We used these observational routinely collected data retrospectively and GPs had no knowledge of an ongoing study. We included five GPCs (rural as well as urban GPCs) covering a large population that reflects the general population in Flanders (Belgium). Only GPCs with complete datasets were included. All GPs of the region are obliged to participate in the OOH care. Therefore, maximal coverage of GPs is guaranteed. In this way, we were able to achieve high validity and completeness of the data.

The use of electronic primary care databases such as iCAREdata can produce valid APQI values [4]. These values were originally developed for primary care, but not specifically for OOH care. Previous research showed minimal differences in antibiotic prescribing between in and out of hours care [4]. The quality of the data depends on the quality of the registrations by the GPs. We suspect that these APQI values give an incomplete picture of antibiotic prescribing. Antibiotics prescriptions linked to a symptom diagnosis, i.e., coding the diagnosis with a code for reasons for encounter such as fever, ear ache, cough, sore throat, etc., were not included in calculating APQI values. However, Flemish GPs not often register symptom diagnoses [7]. Indeed, when looking at the total number of antibiotic prescriptions, only a small percentage was made for symptom diagnosis. Handwritten prescriptions are still possible at the GPC, but are discouraged. Diagnosis shifting is possible as well, i.e., registering an incorrect diagnosis to justify a prescription [8]. Every prescription made by a GP is registered, and no difference is made between an immediate or a delayed prescription in the data. Although delayed prescriptions will add to the quantity of antibiotic prescribing, we know that around 40–60% of patients will collect their antibiotic anyway [31,32,33,34]. We register antibiotic prescribing and not antibiotic consuming or adherence to the prescribed course [31]. 

### 4.3. Implications for Practice and Future Research

Feedback of APQI values could serve as benchmarks of (in)appropriate prescribing. Each GPC could compare their antibiotic prescribing quality with the range of acceptable use to help quantify the magnitude of inappropriate prescribing in OOH primary care. It gives the opportunity to focus on the specific challenges in every organization and to work on prescribing more accurately.

These data could inform future tailored interventions to improve the quality of antibiotic prescribing in OOH primary care, such as we plan to do in the BAbAR project [16], and may also be used to monitor effects of other interventions to improve quality of care at the level of GPCs [35]. However, these indications do not cover all antibiotic prescribing.

## 5. Conclusions

Flemish GPs too often prescribe antibiotics for self-limiting infections in OOH primary care and do not choose the guideline recommended antibiotics. When using disease-specific APQI to define the quality of antibiotic prescribing in OOH care, we suggest to calculate indicator values for two additional disease entities, since a substantial amount of antibiotics are also prescribed for dental abscess and erysipelas. 

## Figures and Tables

**Table 1 antibiotics-08-00079-t001:** Characteristics of the five general practitioner cooperatives (GPCs) providing data for this study.

GPC	Population	Number of GPs	Rural/Urban
Antwerp-East	148,366	112	Urban
Antwerp-North	141,110	119	Urban/rural
Antwerp-Centre	185,358	171	Urban
Tienen	84,430	80	Rural
Zuiderkempen	251,833	270	Rural
Total	811,097	752	

**Table 2 antibiotics-08-00079-t002:** Number of antibiotic prescriptions in a two-year time frame linked to diagnostic group (total patient contacts (excluding home visits) = 111,600).

ICPC-Code	Number of Antibiotic Prescriptions	Percentage
R	11,526	44%
U	3448	13%
S	3125	12%
H	2839	11%
D	1707	6%
missing	1612	6%
Other diagnoses (A, Y, L, X, B, F, W, K, P, N, T)	2179	8%
TOTAL	26,436	100%

ICPC: International Classification of Primary Care; R: respiratory; U: urinary; S: skin; H: ear; D: digestive; A: general and unspecified; Y: Male genital system; L: Musculoskeletal; X: female genital system and breast; B: blood forming organs, lymphatics, spleen; F: eye; W: pregnancy, childbirth, family planning; K: circulatory; P: Psychological; N: neurological; T: endocrine, metabolic and nutritional.

**Table 3 antibiotics-08-00079-t003:** Quality appraisal using disease-specific antibiotic prescribing quality indicators (APQI) complemented with self-generated indicators for erysipelas and dental abscess.

ICPC-Code	Label	% of Patients Prescribed AB	% of Patients Receiving the Guideline Recommended AB	% of Patients Receiving Quinolones
		(<20%)	(>80%)	(<5%)
H71	Acute otitis media	64 [52–71]	74 [72–81]	1 [0–1]
R74	Acute upper respiratory tract infection	30 [15–41]	3 [2–7]	2 [0–2]
R75	Acute sinusitis	51 [39–57]	40 [35–51]	7 [2–9]
R76	Acute tonsillitis	77 [65–87]	6 [3–15]	1 [0–1]
		(<30%)	(>80%)	(<5%)
R78	Acute bronchitis	69 [58–75]	42 [37–54]	11 [5–15]
		(>80%)	(>80%)	(<5%)
R81	Pneumonia	80 [70–85]	46 [42–62]	15 [2–20]
U71	Acute cystitis	91 [77–95]	69 [64–77]	25 [18–32]
		(? > 80%?)	(>80%)	(<5%)
S76	Erysipelas	80 [64–83]	44 [36–59]	1 [0–2]
		(? < 30%?)	(>80%)	(<5%)
D82	Teeth/gum disease	67 [53–72]	42 [38–52]	1 [0–2]

ICPC: International Classification of Primary Care; AB: antibiotics; (): the acceptable range []: variation between the individual GPCs; R78: patients aged between 18 and 75 years with acute bronchitis/bronchiolitis; R74: patients older than one year with acute upper respiratory infection; U71: female patients older than 18 years with cystitis/other urinary infection; R76: patients older than one year with for acute tonsillitis; R75: patients older than 18 years with acute/chronic sinusitis; H71: patients older than two years with acute otitis media/myringitis; R81: patients aged between 18 and 65 years with pneumonia; S76: patients older than one year with erysipelas; D82: patients older than 18 years with teeth/gum disease.

**Table 4 antibiotics-08-00079-t004:** Total antibiotic prescriptions and percentage linked to the diagnosis in a two-year time frame. (total patient contacts (excluding home visits) = 111,600).

ICPC-Code	Label	Total Prescriptions	Percentage
R74 ^†^	Acute upper respiratory tract infection	3564	13%
U71 ^†^	Acute cystitis	3068	12%
R76 ^†^	Acute tonsillitis	2294	9%
R78 ^†^	Acute bronchitis	2223	8%
H71 ^†^	Acute otitis media	2129	8%
missing	Diagnosis missing	1612	6%
S76	Erysipelas	1344	5%
D82	Teeth/gum disease	982	4%
R75 ^†^	Acute sinusitis	965	4%
R81 ^†^	Pneumonia	684	3%
Other diagnoses	Group of diagnoses representing less than 3% each	7571	29%
Total		26,436	100%

^†^ Diagnosis used in antibiotic prescribing quality indicators (APQI).

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
