# Peer review of "Antibiotic Prescribing Quality in Out-of-Hours Primary Care and Critical Appraisal of Disease-Specific Quality Indicators"

_antibiotics, 2019, doi:10.3390/antibiotics8020079_

Round 1
Reviewer 1 Report
The paper is a well written and structured study into antimicrobial prescribing in an out of hours primary care setting in Belgium. This study addresses an important subject area and is a welcome addition to the literature. Some suggestions on minor typographical errors and a few confusing statements are outlined below. I believe that once addressed the manuscript will be a welcome addition to the journal and of interest to the target audience.
General comments
Numbers under ten are interchangeably referred to both as a word (line 97) or numerically (line 98) throughout the paper. This should be consistent.
Specific Comments
Line 37 – Consider changing ‘stays’ to ‘is’ as it would be the meaning of the sentence clearer.
Line 53 – Consider replacing ‘they’ with ‘researchers’. In its present form it sounds colloquial.
Lines 54-55 – Replace ‘but they did not include quality of prescriptions’ with ‘however, this study did not explore the quality of the prescriptions’.
Lines 88-90 – The codes referred to are confusing. The meaning of these codes is clearer as the paper progresses however, it may be worthwhile considered revising these lines to improve the clarity and meaning. For example, specifically defining the codes or referring to where this information may be found later in the paper.
Lines 145-147 – In the absence of quality indicators why were these selected? Is there a reference to support the indicators chosen?
Line 153 – Change ‘around’ to ‘from’.
Line 195 – Should this be ‘DU90%’ as referred to previously (line 115)? If not, what is being referred to by ‘D90%’?
Line 210 – ‘Prescription’ should be plural ‘prescriptions’.
Line 211-213 – The wording is confusing and the English needs revising. A suggested revision is ‘For nearly all indications the recommended type of antibiotics, according to the guidelines, are not prescribed apart from for otitis media which is almost within the accepted range’.
Line 217 – Consider adding ‘rate’ after ‘prescribing’.
Line 233 – Consider adding ‘the’ before ‘use’ to make the meaning clearer.
Lines 245-246 - The wording is confusing and the English needs revising.
Line 267 – Should ‘withheld’ be ‘included’. Were these data not described in the paper? Is it not GPCs with incomplete data which were excluded? Review this.
Line 277 – Replace ‘to’ with ‘in’.
Line 278 – Delete ‘few’ and ‘diagnosis’ should be plural (‘diagnoses’).
Author Response
Dear Reviewer,
Thank you for reviewing our manuscript: “Antibiotic prescribing quality in out-of-hours primary care and critical appraisal of disease-specific quality indicators.”
We are delighted to have the opportunity to revise the manuscript in light of the points and concerns that were brought up by the reviewing process.
We believe we have addressed all the comments as itemised below:
General comments
Numbers under ten are interchangeably referred to both as a word (line 97) or numerically (line 98) throughout the paper. This should be consistent.
Thank you for this comment. We used words throughout the paper to indicate numbers under ten, unless referring to tables/figures or percentages.
Specific Comments
Line 37 – Consider changing ‘stays’ to ‘is’ as it would be the meaning of the sentence clearer.
Thank you for this comment. We changed ‘stays’ to ‘is’
Line 53 – Consider replacing ‘they’ with ‘researchers’. In its present form it sounds colloquial.
Thank you for this comment. We replaced ‘they’ with ‘researchers’
Lines 54-55 – Replace ‘but they did not include quality of prescriptions’ with ‘however, this study did not explore the quality of the prescriptions’.
Thank you for this comment. We followed the suggestion and replaced this sentence with: ‘however, this study did not explore the quality of the prescriptions’.
Lines 88-90 – The codes referred to are confusing. The meaning of these codes is clearer as the paper progresses however, it may be worthwhile considered revising these lines to improve the clarity and meaning. For example, specifically defining the codes or referring to where this information may be found later in the paper.
Thank you for this comment. We revised the lines as follows: The software can differ depending on the GPC, ,but diagnoses are always selected using Thesaurus terms in the Belgian Bilingual Biclassified Thesaurus (3BT) By selecting a diagnosis in 3BT, an International Classification of Primary Care (ICPC-2-R) code is automatically linked. [19] Prescriptions are registered using the database of the Belgian Centre for Pharmacotherapeutic Information[20], which is linked to the Anatomical Therapeutic Chemical (ATC) classification system [21,22].
Lines 145-147 – In the absence of quality indicators why were these selected? Is there a reference to support the indicators chosen?
Thank you for this comment. These two indicators were chosen based on the number of prescriptions made for these two diagnoses. The limitations we proposed are based on the local guidelines and consensus within our research team.
A review from Saust et al. evaluated existing quality indicators, they identified 130 QIs for diagnosis and antibiotic treatment of infectious diseases, with only 3 for skin infections, that only evaluate the choice of antibiotics and non for dental infections.
Ref: Saust LT, Monrad RN, Hansen MP, Arpi M, Bjerrum L. Quality assessment of diagnosis and antibiotic treatment of infectious diseases in primary care: a systematic review of quality indicators. Scand J Prim Health Care. ;34(3):258–266. doi:10.1080/02813432.2016.1207143
We added the reference to the paper.
Line 153 – Change ‘around’ to ‘from’.
We changed ‘around’ to ‘from’.
Line 195 – Should this be ‘DU90%’ as referred to previously (line 115)? If not, what is being referred to by ‘D90%’?
We corrected ‘D90%’ to ‘DU90%’
Line 210 – ‘Prescription’ should be plural ‘prescriptions’.
We corrected it to the plural form.
Line 211-213 – The wording is confusing and the English needs revising. A suggested revision is ‘For nearly all indications the recommended type of antibiotics, according to the guidelines, are not prescribed apart from for otitis media which is almost within the accepted range’.
Thank you. We used your suggestions to improve this sentence.
Line 217 – Consider adding ‘rate’ after ‘prescribing’.
We added ‘rate’ after ‘prescribing’
Line 233 – Consider adding ‘the’ before ‘use’ to make the meaning clearer.
We added ‘the’ before ‘use’
Lines 245-246 - The wording is confusing and the English needs revising.
We replaced this sentence and changed it to:“Dolk et al. found that the majority of antibiotic prescriptions in English primary care were for infections of the respiratory and urinary tracts, followed by skin infections and only a small number for dental problems.”
Line 267 – Should ‘withheld’ be ‘included’. Were these data not described in the paper? Is it not GPCs with incomplete data which were excluded? Review this.
Thank you for this comment. Indeed it should be ‘included’. These are indeed the data we included in the paper. The iCAREdata database holds data from more than 5 GPCs.
Line 277 – Replace ‘to’ with ‘in’.
We replaced ‘to’ with ‘in’.
Line 278 – Delete ‘few’ and ‘diagnosis’ should be plural (‘diagnoses’).
Thank you for this comment. We changed the sentence to: “However Flemish GPs not often register symptom diagnoses”
Reviewer 2 Report
This is a very well done observational study of a large community cohort examining the important question of inappropriate antibiotic use in primary care, focusing on after-hours prescribing in an urgent care setting. Not surprisingly, the authors find a high degree of inappropriate antibiotic prescribing, especially with regards to use of fluoroquinolones for urinary tract infection and antibiotics overall for URI and acute bronchitis. The findings are hardly new - excessive antibiotic use in primary care has been well documented, but the results can certainly be useful for policy development and quality of care improvement.
Specific issues to fix: The table describing antibiotic use is way too complicated - the headings require a code to decipher, which is awkward and needs fixing. Make the headings self-explanatory and eliminate the A,B,C headings you currently have.
Author Response
Dear Reviewer,
Thank you for reviewing our manuscript: “Antibiotic prescribing quality in out-of-hours primary care and critical appraisal of disease-specific quality indicators.”
We are delighted to have the opportunity to revise the manuscript in light of the points and concerns that were brought up by the reviewing process.
We believe we have addressed the comments:
Specific issues to fix: The table describing antibiotic use is way too complicated - the headings require a code to decipher, which is awkward and needs fixing. Make the headings self-explanatory and eliminate the A,B,C headings you currently have.
Thank you for this comment. We eliminated the ABC headings. We deleted the column APQI n°. And we deleted some unnecessary text/lines in the caption.